# Wood-based superblack

Bin Zhao[1], Xuetong Shi[2], Sergei Khakalo [3,4], Yang Meng[5], Arttu Miettinen [6], Tuomas Turpeinen[7], Shuyi Mi[8], Zhipei Sun [8,9], Alexey Khakalo[10], Orlando J. Rojas [1,2] ✉ & Bruno D. Mattos[1] ✉

Light is a powerful and sustainable resource, but it can be detrimental to the performance and longevity of optical devices. Materials with near-zero light reflectance, i.e. superblack materials, are sought to improve the performance of several light-centered technologies. Here we report a simple top-down strategy, guided by computational methods, to develop robust superblack materials following metal-free wood delignification and carbonization (1500 °C). Subwavelength severed cells evolve under shrinkage stresses, yielding vertically aligned carbon microfiber arrays with a thickness of ~100 μm and light reflectance as low as 0.36% and independent of the incidence angle. The formation of such structures is rationalized based on delignification method, lignin content, carbonization temperature and wood density. Moreover, our measurements indicate a laser beam reflectivity lower than commercial light stoppers in current use. Overall, the wood-based superblack material is introduced as a mechanically robust surrogate for microfabricated carbon nanotube arrays.

Materials displaying ultralow light reflectance, known as superblack materials, are in high demand for advanced light harvesting and management, such as blackbody cavity coatings[1], solar energy harvesters[2,3], and pyroelectric detectors[4]. Superblack materials are used as coatings for baffles in optical instruments, including telescopes, e.g., to suppress undesirable light scattering[5]. In the animal kingdom, superblack features have been used for various reasons including camouflage[6], thermoregulation[7], as well as to enhance conspicuousness[8–10] and aposematism[11]. Superblack is typically obtained by combining chemically derived blackness (arising from moieties with short optical bandgaps) with structural features that induce multiple internal light reflections and show low backscattering[12–15].

Several efforts have attempted to recreate the ordered ultrastructures found in living superblack organisms (e.g., deep-sea fish[6], butterfly[10], paradise birds[8], peacock spiders[9] and inexpectata beetles[16]). Earlier mention to superblack materials relates to the development of textured surfaces by etching electroless nickel-phosphorus deposits with oxidizing acid solutions. These black metallic microcavity surfaces displayed light reflectance of ca. 0.2–0.5%[5,15]. Arrays of vertically aligned carbon nanotubes (CNT) have been developed with light reflectance below 0.03%[17,18], an order of magnitude lower than earlier Ni-P coatings. Aligned CNT arrays are grown from catalyst-containing metallic substrates by chemical vapor deposition, a method that is often described as energy-demanding and environmentally sensitive[12,13,17]. Superblack features could also be achieved in isotropic carbon aerogels of extremely low density[19–21]. These carbon constructs, although displaying ultralow light reflectance, lack cohesion[19,22] and are fragile[12,13,23,24], thus imposing limitations for their application. More recently, hemispherical light reflectance levels have been lowered down to 0.02% by embedding a light absorbing polymer into a microcavity textured surface[14]. The

[1]Department of Bioproducts and Biosystems, School of Chemical Engineering, Aalto University, Espoo FI-02150, Finland. [2]Bioproduct Institute, The University of British Columbia, Vancouver, BC V6T 1Z3, Canada. [3]Department of Civil Engineering, School of Engineering, Aalto University, Espoo FI-02150, Finland. [4]Integrated Computational Materials Engineering, VTT Technical Research Centre of Finland Ltd, Espoo FI-02044, Finland. [5]Faculty of Chemical Engineering, Kunming University of Science and Technology, Kunming 650500, PR China. [6]Department of Physics, University of Jyvaskyla, Jyväskylä FI-40014, Finland. [7]Fiber Web Processes, VTT Technical Research Centre of Finland Ltd, Jyväskylä FI-40400, Finland. [8]Department of Electronics and Nanoengineering, Aalto University, Espoo FI-02150, Finland. [9]QTF Centre of Excellence, Department of Applied Physics, Aalto University, Espoo FI-02150, Finland. [10]Cellulose Coatings and Films, VTT Technical Research Centre of Finland Ltd, Espoo FI-02044, Finland. ✉e-mail: orlando.rojas@ubc.ca; bruno.mattos@aalto.fi

resulting material overcomes problems related to cohesion and fragility, but it still relies on synthetic precursors. Noticeably, the differentiation between black and superblack is somewhat arbitrary and in this paper we refer to superblack materials as those having a reflectance <0.4%.

The examples of living superblack entities and the success of the seminal research on manmade superblack systems have incentivized the development of light absorbing materials from renewable precursors, while aiming at increased robustness and lowered processing demands. On that note, researchers have used wood as a biobased ultrastructured precursor to develop materials with antireflective features. Generically, the cellular ultrastructure in wood can be described as a construct of vertically aligned hollow tubes. While the tubular arrangement of (carbonized) wood can induce light entrapping mechanisms, its anisotropic cell structure is notable for maximizing stiffness and strength at a remarkably low density[23]. The axial strength along the tree-growth direction is attributed to vertically aligned cellulose elementary fibrils embedded in a matrix of hemicelluloses and lignin[25]. Given its structural strength, wood retains its tubular meso-scaled features after carbonization, which have led to surfaces with light absorption ranging from 97–99%[2,26–28]. Antireflective materials from wood have been made using bulk modification, compositing with light-absorbing particles[29,30], and surface treatments[28,31]. Compared to CNT arrays and microcavity textured coatings, the light entrapping features developed with wood-based black materials are of a larger dimensional spacing, leading to light reflectance in the 1–3% range.

In this work we propose a method that leads to near full light absorption (~99.65%) by taking advantage of wood's inherent features, and a vertically aligned cellular structure that is formed as a hierarchical assembly of elementary cellulose microfibrils. By using simple deconstruction methods, the tubular structure in wood is converted into microfiber arrays that strengthen multiple internal light reflections and lower backscattering. In the context of superblack materials, wood is proposed to bring positive prospects as far as sustainability[32] and carbon footprint compared with current light management technologies that are based on metals[15] or synthetic precursors[18–20]. Our top-down approach starts with wood delignification (Fig. 1a, b) to isolate cellulose microfibrils in the cell walls, followed by carbonization (Fig. 1c). The main strategy relies on the manipulation of the cellulose microfibril assembly across the cell wall so that they become disconnected and split during carbonization, but to an extent that they maintain a high-level of mechanical integrity. This process generates a vertically aligned microfiber array that is in a subwavelength size scale (Fig. 1c), which enhance multiple internal light reflections and displays low backscattering (Fig. 1d), even under intense light illumination (Fig. 1e). We achieve light reflectance <0.4%, being an order of magnitude lower than that of simply carbonized wood (Fig. 1f).

## Results

### Interplay between light interactions and carbonized wood anatomy

Finite element (FE) modelling of optical properties was employed to identify the key wood anatomical elements that affect light interactions. The results were then used to guide the fabrication of wood-based superblack materials (Supplementary Figs. 1–6). Experimentally, we considered a low-density wood specie, balsa, which comprises vessel elements (3–9%), ray cells (20–25%) and fibers (66–76%) (Fig. 2a and Supplementary Fig. 7)[33,34]. Except for the expected shrinkage, the overall morphology of balsa maintained upon carbonization at ~1100 °C (Fig. 2b, c). The dimensions of single wood cells were determined by high-resolution 3D reconstruction (nano/micro-tomography, Fig. 2d and Supplementary Fig. 7) and the data was used to create 2D cell structures for FE modelling (Fig. 2e). The effects on light reflection were examined systematically considering cell wall thickness, lumen width and cell end tilt (tapering versus blunt ends, defined by the respective slope) as well as fiber length (Fig. 2f, j). The FE

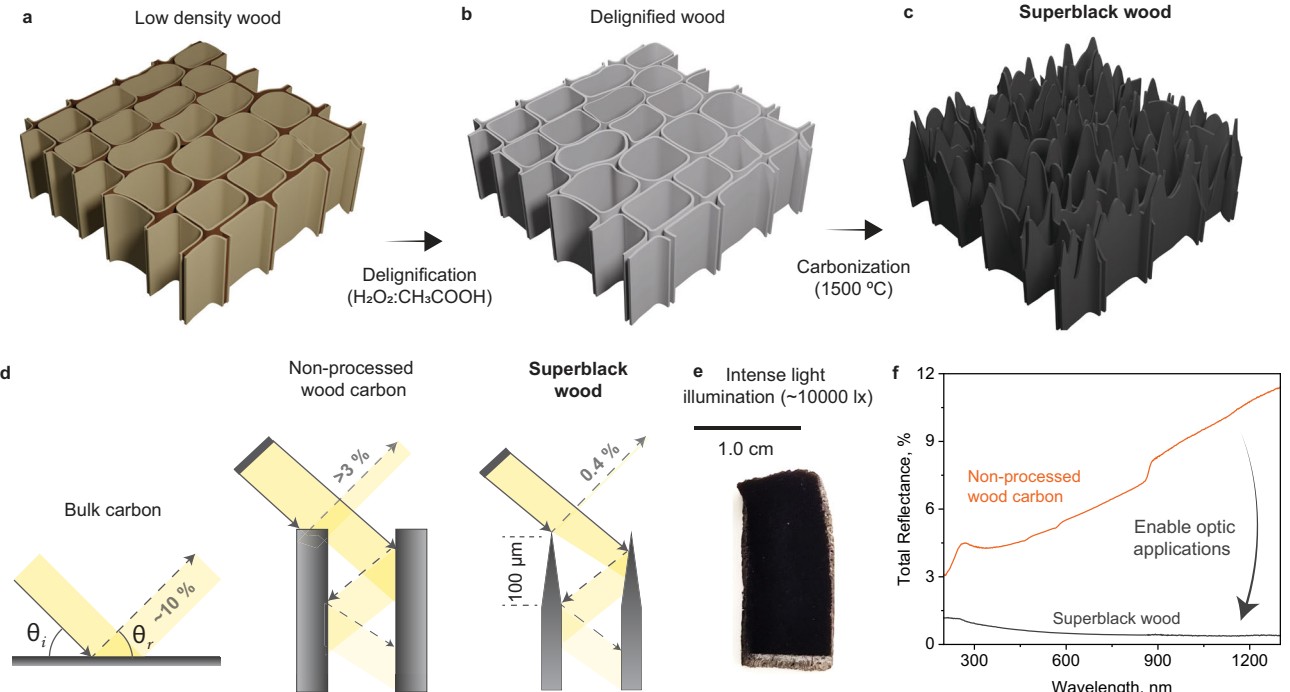

**Fig. 1 | Fabrication of superblack wood. a–c** Schematic illustration showing the structure of low-density wood (W), delignified wood (DW) and our superblack wood. **d** Schematic illustration showing the light reflection in bulk carbon, a cross-section of wood carbonized below 1100 °C, and superblack wood. **e** Our superblack wood surface under intense light illumination (~10,000 lx). The side surface gets whiteout, in contrast to the low reflection of the superblack wood cross section. **f** Measured light reflectance of non-processed wood carbon and our superblack wood.

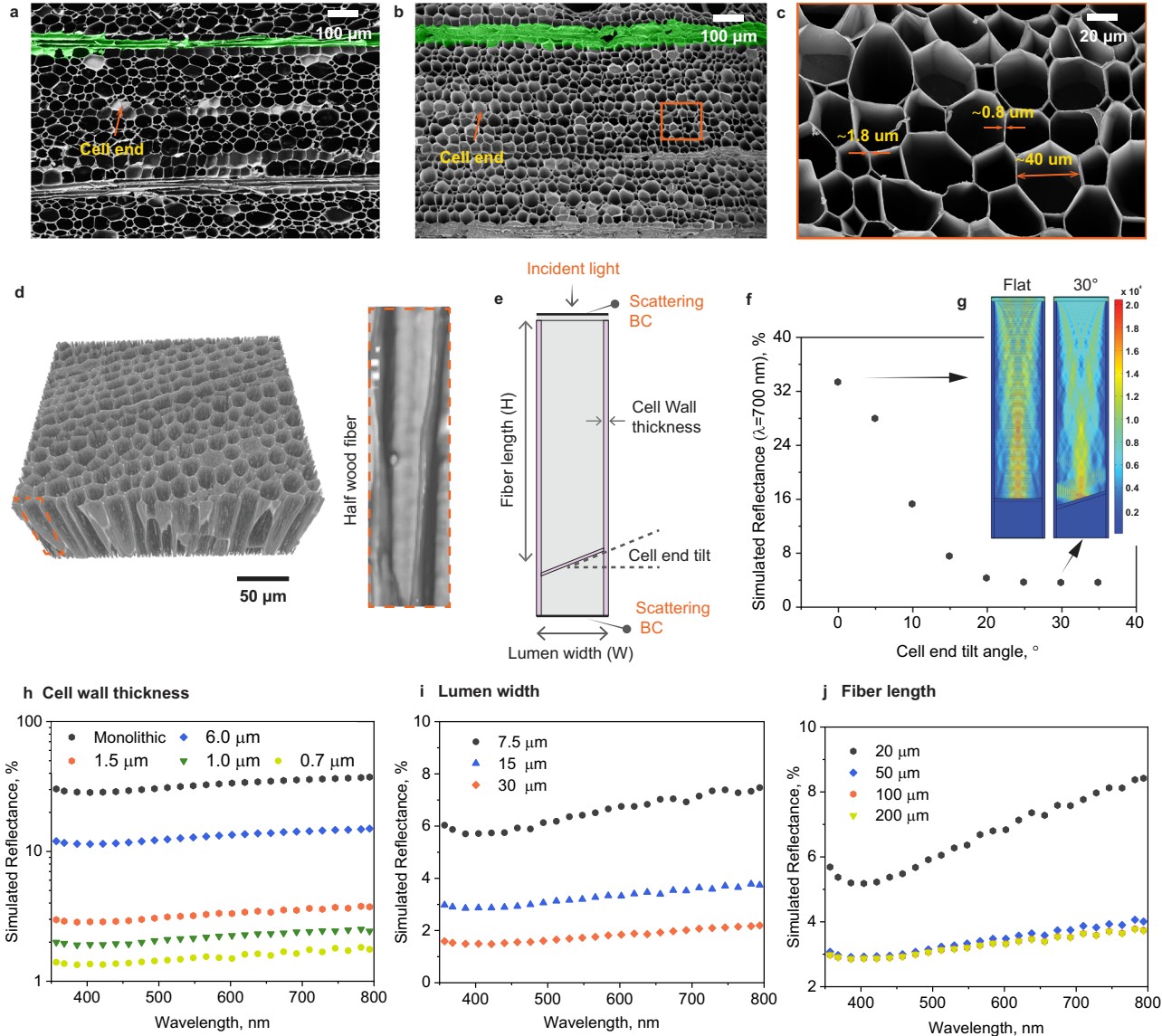

**Fig. 2 | Morphology of traditional carbonized wood and FE simulation on their effect on light interactions.** Scanning electron microscope (SEM) images of cross section of (**a**) wood (W) and (**b**, **c**) carbonized wood at 1100 °C, cW. **a**, **b** The ray cells (shown in green color) are collapsed into a nearly monolithic structure. **d** 3D visualization of microCT scans of carbonized balsa wood. **e** Idealized procedure adopted to investigate carbonized wood-light interactions by FE modelling (BC: boundary conditions). Simulated light reflectance on (**f**) 2D vertical cylinder array with varied cell end tilt angle (**g**) displaying the distribution of the electric field norm for cell end tilt angle at 0° and 30°, as well as (**h**) cell wall thickness, (**i**) lumen width, and (**j**) fiber length.

simulations do not account for the effect of pit elements at fiber-fiber lateral connections, which are abundant in balsa. Pits are suggested to induce light propagation across wood lumina, likely increasing the light pathlength across cell elements and therefore the effect on reflectivity was minor.

Wood cross-section mainly consists of solid cell walls (fiber walls) and vertical cellular voids (lumina)[35]. The vertically aligned cell structures enable multiple internal light reflections, as shown by the calculated (FE) reflectance of ~3%, down from ~15% predicted for a flat, featureless carbon structure (Fig. 1d and Supplementary Figs. 1 and 2). FE simulation shows that an increased tilt of cell ends enhances light absorption compared to that of lumina with blunt ends (as shown in the distribution of the electric field norm, Fig. 2g). Figure 2f shows the reduction in light reflectance for larger tilts in cell ends (up to 30°). The cell end tilt enhances light absorption by providing greater distance for light propagation inside the wood lumina. In wood, such tilt varies from 10 to 80°[34]; for simplicity and as a reference, we assumed a tilt at

30° to investigate the effect of other wood anatomical elements on the light-matter interactions.

Light reflectance positively correlates with fiber wall thickness (Fig. 2h and Supplementary Figs. 1–3). Wood with thinner fiber walls (lower filling fraction) reduce the refractive index (RI) inhomogeneities of across the wood, so as the light scattering according to Maxwell-Garnett approximation[13,36,37]. In addition, a thinner fiber wall results in Mie scattering down-scaling to the object dimensions[38,39]. A larger lumen width also favors light trapping due to the consequently lower volume-filling of the solid cell walls that are defined as light scatters (Fig. 2i and Supplementary Figs. 2, 3).

The fiber length should be greater than 50 μm to induce effective multiple internal light reflections and confine the incident light in the assumed cylindrical arrays (Fig. 2j, Supplementary Figs. 3i and 4). In balsa wood, the fiber length is up to 400 and 800 μm, respectively, considering the typical maximum length of vessel and fiber elements (in FE models simplified as fiber length)[33,34], thus greatly favoring

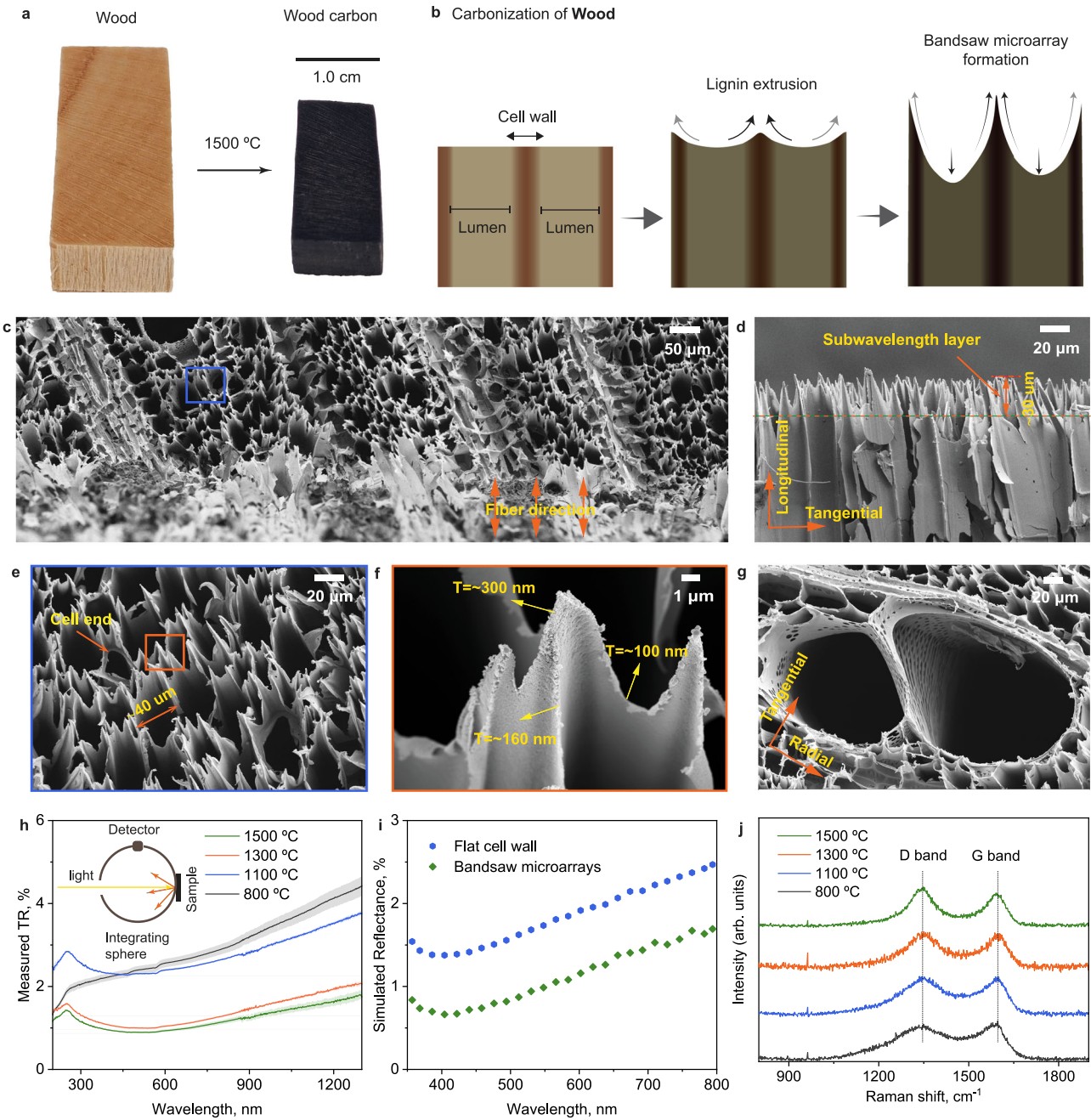

**Fig. 3 | Formation of bandsaw-like carbon structures in wood. a** Photographs showing wood (W) and carbonized wood (cW) at 1500 °C. **b** Proposed mechanism for the formation of bandsaw-like microarrays in cW. **c–g** SEM images of the morphology of cW obtained at 1500 °C. **c** SEM image showing bandsaw-like microarrays were formed in wood cell walls, while ray cells were converted into flat structures. **d** Side-view SEM image showing vertically aligned bandsaw-like microarrays in wood. **e, f** Top-view SEM image showing the morphology and dimension of bandsaw-like microarrays. **g** Top-view SEM image showing carbonized vessels and ray cells. **h** Measured total reflectance (TR) of cW obtained at 800–1500 °C. **i** Simulated light reflectance of cW with flat cell walls and bandsaw-like microarrays. **j** Raman spectra of cW obtained at 800–1500 °C.

multiple internal light reflections. Near-surface cell ends displaying on wood cross-sections (Fig. 2a) clearly indicated lower light absorption by FE modelling. After carbonization, the ray cells (ca. 30 μm in length[33]) are collapsed into a nearly monolithic structures (Fig. 2a, b, shown in green color), where light reflection is intensified.

## Lowering the light reflectance of carbonized wood

Following the above simulation results, here we discuss how we obtained carbonized wood with low backscattering and reflection (<1%). Current wood carbons obtained via carbonization at temperatures below 1100 °C led to preserved wood microstructures and a

cross-section reflectivity of ca. 3%[2,26,27]. Herein, we started manipulating the wood microstructure by elevating the carbonization temperature to 1300 °C (Fig. 3a). The resultant carbonized wood (cW) is visually much blacker than wood carbonized at temperatures below 1100 °C. Using scanning electron microscopy (SEM) we examined the microstructures of the cW samples to better understand their morphology-reflectance relationships and correlate the light-matter interaction. We observed microstructures in the form of bandsaw-like microarrays (Fig. 3b–f). The sharp edges of the latter system were always located at the cell corners and were aligned along the grain direction, with an approximate height of ~30 μm (Fig. 3d). The cell

corners were only 300-nm thick, while the dimensional length scales were as low as 100 nm at the middle lamella between cell corners (Fig. 3f). The thickness distribution of bandsaw-like microarrays in the cW precisely followed the dimensions of wood cell wall configurations, thinner in the middle lamella and thicker in the cell corners (Supplementary Fig. 8)[40–42]. Moreover, cell ends were shown to be partially degraded upon carbonization, thus better inducing multiple internal light reflections (Fig. 3e). After carbonization at 1300 °C, the ray cells remained in the form of flat structures (even after carbonization at 1500 °C, Fig. 3c, g). Note that the carbonized ray cells are responsible for the light stripes that become clearly visible under intense light illumination (-10,000 lx) (Supplementary Fig. 9 and detailed discussion), which were invisible under room light illumination (-500 lx) (Fig. 3a).

The subwavelength structures of cW lowered the light reflectance, from ca. 3% (typical carbonized wood) to 0.9% (Fig. 3h) when we elevated the carbonization temperature from 1100 to 1300 °C. This is a result of the bandsaw structures formed at 1300 °C (Fig. 3d compared to Fig. 2b, c). FE simulation confirmed the effects of sharp edges on the substrate-light interactions (Fig. 3i and Supplementary Fig. 5), where the simulated reflection decreased from an initial value of 2% to <1%. The formed submicron arrays with graded refractive index scattered less incident light[36], reducing the total light reflection.

We note that chemical changes can improve the blackness of wood carbons, related to differences in graphitization and associated optical bandgaps[43,44]. However, the light reflectance of cW radial plane was >3% at carbonization temperature of 1500 °C (Supplementary Figs. 10–11), far above that measured for the cross-section (-0.9%). The Raman spectra of cW obtained from carbonization from 800 to 1500 °C confirmed that the resulting wood carbon are chemically similar as far as graphitization degree (Fig. 3j)[45,46]. The intensity ratio between D-band (1330 cm$^{-1}$, disordered graphite) and G-band (1590 cm$^{-1}$, ideal graphite crystal lattice) increased minimally with the carbonization temperature. Therefore, the boosted light absorption performance (from -97% to -99.1%) of cW (at >1300 °C) was mainly ascribed to physical changes rather than chemical ones, i.e., the bandsaw-like microstructures.

### Fabrication of wood-based superblack material

Following carbonization at 1500 °C, wood cell walls were converted into subwavelength structures, and the light reflectance dropped to -0.9%, not yet at a superblack level (-0.4% as considered in this paper)[15]. Hence, we targeted subwavelength structures of finer dimensions, expected to further reduce backscattering and reflectance, for example, by wood cell wall deconstruction through delignification (using $H_2O_2$:$CH_3COOH$) prior to carbonization. Delignification is known to partially isolate wood microfibrils in the cell walls[47,48] and maintains a cellular morphology containing a loosened fibrillar structure (Supplementary Fig. 8b), which facilitates structural deconstruction during carbonization. Compared to the cW, the cross-section surface became blacker when delignified wood (DW) was carbonized at -1500 °C (cDW) (Fig. 4a). In fact, the light reflectance of cDW was lowered to -0.36% (Fig. 4c), reaching superblack levels[15].

Near-complete removal of lignin, and associated cellulose microfibril isolation, was required to achieve superblack levels (Fig. 4i). We investigated the effect of microfibril bundle size on the respective reflectance of the material by using FE methods. We show that a structure comprising porous and sparse microfibril assemblies significantly lowered light reflection (Fig. 4j and Supplementary Fig. 6). SEM revealed that the cross section of cDW featured microfiber arrays whose primary units displayed sharp tips (tens of nanometers in dimension), which are subwavelength structures that depart from the typical wood anatomical elements (Fig. 4e, h). The wavelength range in which the reflection reached a minimum shifted to the near infrared (NIR) range (>700 nm) (Fig. 4c). FE modelling indicated that light-

matter interaction was suppressed in the NIR range when cylindrical arrays were replaced by fibrillar arrays with sharp tips (Fig. 4j). The lignin content of DW should be below 3% to loosen the microfibril in cell walls, and to allow their conversion into microfiber carbon arrays. Although maintaining lignin improves the carbon yield (higher carbon content[45]), a residual lignin content over 3% would limit the cell wall disintegration and impair the level of achievable blackness (Supplementary Figs. 12 and 13).

The structural evolution that takes place upon the carbonization of delignified wood was investigated in detail. We found that loosened cellulosic fibers (Supplementary Fig. 8) were torn into vertically aligned microfiber arrays (Indigo color in Fig. 4d, e). Fibers composed of isolated microfibrils after removal of lignin, were torn into multiple microfibers under the build-up of stresses that occurred during carbonization and ensuing multidirectional shrinkage (Fig. 4b). Carbonization at 1100 °C formed distorted carbon microfibers (Supplementary Fig. 14). By contrast, carbonization at >1300 °C led to vertically aligned microfibers. The ray cells, remained intact but were weakened during delignification, becoming separated from the adjacent elements, and shrunk significantly upon carbonization (Fig. 4d, e, shown in green color). The ray cells (width of the order of tens of microns) were converted into ca. 1 μm-wide thin bands (Fig. 4e), which led to the disappearance of the light stripes even under intense light illumination (-10,000 lx) (Supplementary Fig. 9). It was also noted that near-surface cell ends, and ray cells were rarely observed in the 3D reconstruction of cDW (Supplementary Fig. 7). Carbonized vessels shrunk and deformed slightly, but they kept the original morphology of DW (Fig. 4d, e, shown in cyan color).

Balsa is a light-weight wood whose density values typically range between 100 and 250 kg/m$^3$ [33]. Balsa wood density plays a major role during delignification, especially to reach lignin content <3%. While near-complete delignification takes 6 h for dense balsa (>340 kg/m$^3$), it takes only 2 h for a light balsa (<90 kg/m$^3$). The formation of the carbon microfiber arrays in cDW was independent of the wood density (from 90 to 340 kg/m$^3$) (Supplementary Figs. 15–22); however, wood density had an impact on the dimensional features of the resulting carbon microfiber arrays (Supplementary Figs. 15–23). A higher reflectance (-0.64% and 0.61%) was observed in cDW obtained from light (<90 kg/m$^3$) and dense wood (>340 kg/m$^3$), respectively (Fig. 4k). The lowest reflectance was found at a wood density of 160 kg/m$^3$. Carbons fabricated from delignified pine and cedar wood show a minimum light reflectance of -0.77% and -0.85%, respectively. Such values, larger compared to that of balsa, mainly are attributed to the thicker cell wall and higher wood density (Supplementary Fig. 23). Carbonized delignified wood carried the effect of precursor specie and density on the developed blackness (Supplementary Figs. 15–23). Importantly, the formation of (black) carbon microfiber array following our approach is not limited to balsa wood but is applicable to other wood species, such as pine (Supplementary Fig. 23).

### Practical implications of superblack wood

Superblack wood showed low light transmittance through the longitudinal direction (Fig. 5a). Total light reflection and transmittance of cDW (thickness of 3 mm) were <0.4% (Fig. 4f) and <0.02% (Fig. 5a), respectively. The transmittance signal of 7-mm samples was below the detector noise level. Hence, a near-perfect light absorption system was demonstrated for superblack wood. Long wood fibers (ca. 800 μm) with cell ends (fiber-fiber axial connections)[33], prevented the incident light from passing through the superblack wood. Superblack wood was antireflective and opaque to visible light. This contrasts with CNT-based technologies comprising sparse assemblies of hollow nanotubes with limited height (<500 μm). Therefore, the transmittance level of vertically aligned CNT arrays (VACNT) has been addressed by the choice of underlying substrate, e.g., silicon wafer from which CNTs are grown[1,12].

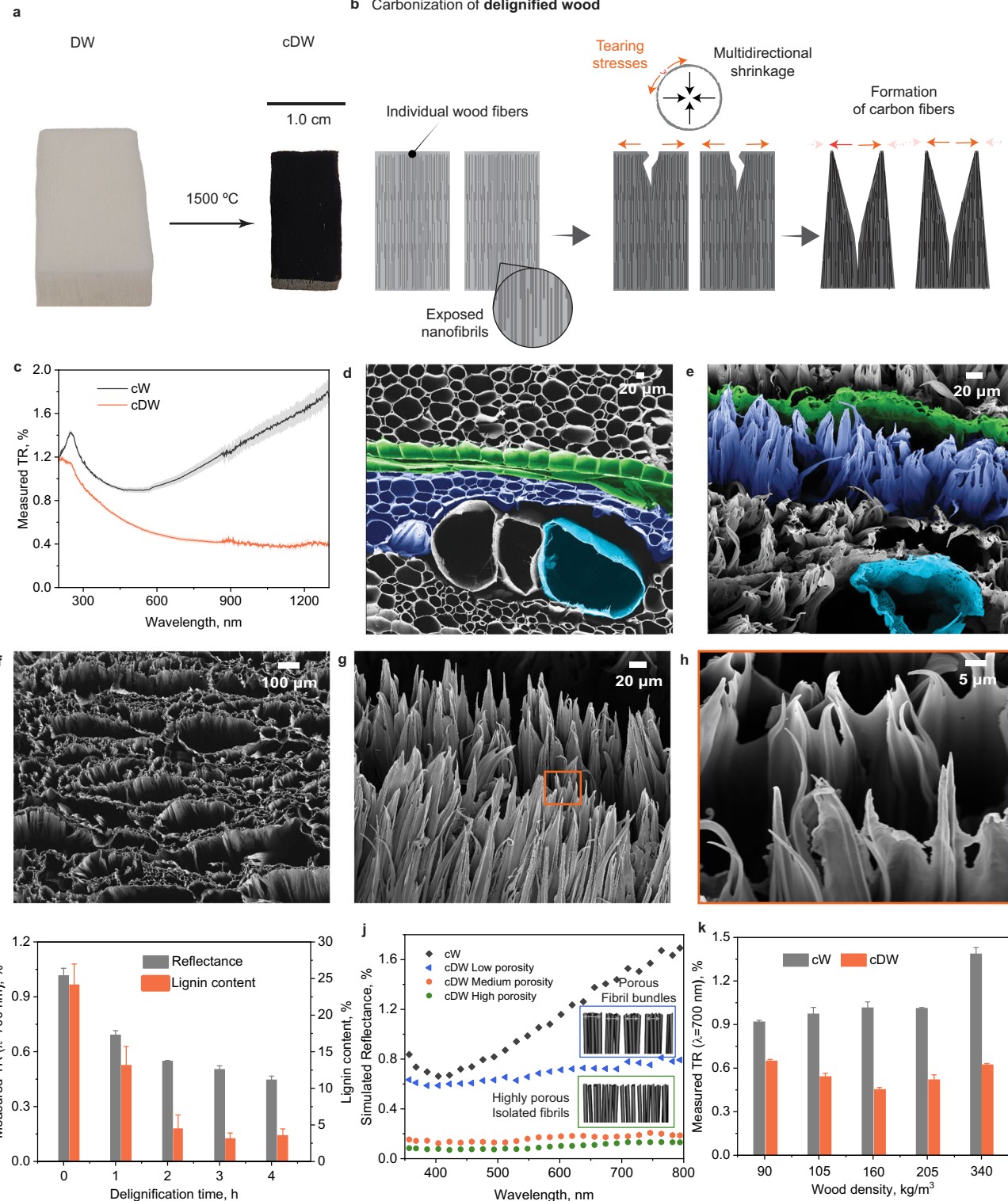

**Fig. 4 | Formation of sharp microfiber carbon array in delignified wood.**
**a** Photographs showing delignified wood (DW) and DW carbonized at 1500 °C
(cDW). **b** Mechanism for the formation of vertically aligned microfiber arrays in
cDW. **c** Total light reflectance of cW and cDW. SEM images of (**d**) DW and (**e**) cDW
obtained at 1500 °C. Ray cells, lumens and vessels were colored in green, indigo,
and cyan, respectively. **f** Top-view SEM image showing the cross-section of DW
carbonized at 1500 °C. **g**, **h** Side-view SEM images showing the morphology and
dimension of vertically aligned carbon microfiber arrays. **i** Total light reflectance (at
$\lambda = 700$ nm) of cDW and lignin content of DW as a function of delignification time.
**j** Simulated light reflectance of cW compared to cDW microfiber arrays with varied
inter-fiber porosity. **k** Total light reflectance (TR, at $\lambda = 700$ nm) of cW and cDW as a
function of balsa wood density.

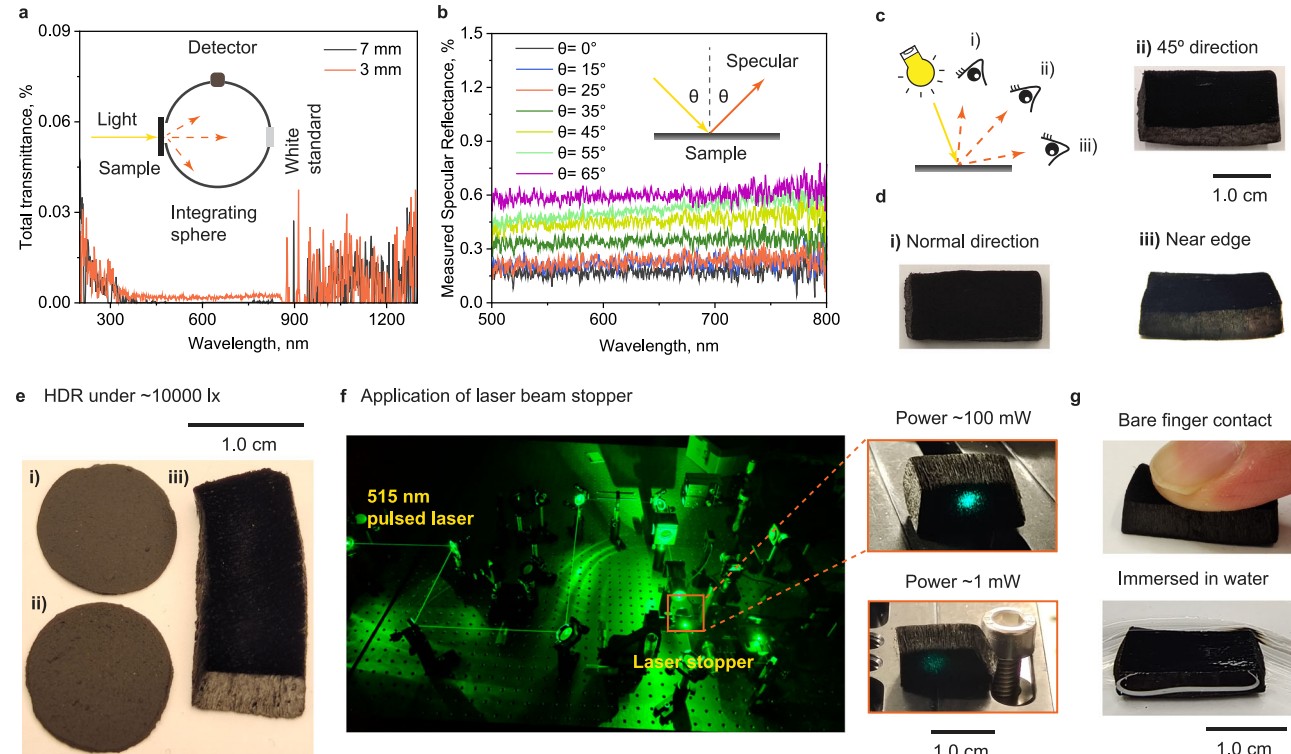

**Fig. 5 | Opacity, angle dependency and practical applications of superblack wood. a** Total transmittance of cDW through the longitudinal direction (tree-growth direction) with thicknesses of ~3 and 7 mm. **b** Specular reflectance of cDW using illumination from 0 to 65° with respect to the normal direction. **c**, **d** Directional reflection properties of cDW using normal illumination and observation at 90° (i), 45° (ii) and near-edge ~0° (iii). **e** Film of carbon nanoparticles with diameter of (i) 900 nm (3% reflection)[50], (ii) 200 nm (4% reflection)[50], and (iii) Superblack wood photographed under intense light illumination (~10,000 lx) in high dynamic range (HDR). **f** Low light scattering from superblack wood under laser (power of 100 and 1 mW). **g** Photographs of superblack wood with bare finger contact and immersed in water.

Light reflectance under oblique incident light is of practical importance. Superblack wood maintained a specular reflectance of <0.5% at incident angles as high as 55° (Fig. 5b). Photographs in several directions supported the low angle dependence of superblack wood (Fig. 5c, d) and showed that a larger incident angle resulted in less light trapping efficiency (light approaches perpendicularity to some misaligned microfibers). The low reflectivity of superblack wood was independent of the incident light angle. This is in contrast with VACNT superblack, which is shiny when viewed by the edge due to the strong Fresnel surface reflection[49]. The surface of superblack wood remained significantly dark under intense light illumination (~10,000 lx) (Fig. 5e). Note: As reference, we tested two films of carbon nanospheres (with light reflectance of 3–4%)[50]. The angle-independent antireflective and opacity features of the developed superblack wood prompted testing for applications such as laser beam trapping (Fig. 5f). Under strong laser illumination (100 mW), very low scattered reflection (~30 ppm) was observed with superblack wood, which is lower than laser beam block (LB1/M) and is 3 times lower than laser beam block (LB2/M) (Supplementary Fig. 24 and Supplementary Table 1). Using a laser power commonly used for experimental laser alignment (1 mW), the laser spot on cDW declined significantly, demonstrating excellent performance as laser beam stopper. Using medium laser illumination (12 mW), the power of laser beam transmitted through superblack wood along the longitudinal direction is <0.1%, indicating excellent opacity.

The mechanical robustness of superblack materials is an important factor for their implementation in real life. Superblack materials are typically achieved with structures of low density or low filling factor[12,13]. VACNT-based materials are fragile by nature as the adhesion of the CNT array to the underlying substrate is driven by weak secondary interactions. Moreover, the moisture sensitiveness of VACNT could potentially lead to loss of blackness[13,14]. Supreme black[14] and Ni-P coatings[5] are touch-proof and more robust than VACNT materials. Carbon aerogels can also be considered touch-proof, depending on the use of binders[19,23]. Superblack flock sheets (based on rayon fibers at 40 wt.%, bound by acrylic and polyamide resins at 60 wt.%) are highly robust, although relatively dense (>350 kg/m³) and are described as a flammable (MSDS of Visible Light Absorbing Flock Sheet)[51]. Living superblack structures in nature tend to be quite robust given the associated evolution and adaptation processes. An example is the deep-sea fish scales, where melanosomes have size and aspect ratio optimized to yield strength[6]. Biological structures could, in fact, inspire the design of robust synthetic superblack materials.

Common black wood-based carbons show a compressive strength over 450 kPa (Supplementary Fig. 25)[2,26]. Following the deconstruction of the wood structure into microfiber arrays, the superblack wood was still mechanically robust, showing compressive strength of 140 kPa (Supplementary Fig. 25). Superblack wood withstands mild water washing (no stirring) and oven drying at 105 °C (Fig. 5g). Six-months storage in a conditioned environment (23 °C and 50% of relative humidity), compressed air blowing (under ~50 kPa), tape peeling and bare finger touch (under ~5 kPa) produced no damage in the developed superblack wood (Supplementary Fig. 26). Sandpaper linear abrasion (P2000 under ~7 kPa) and compressive strength test clearly damaged the superblackness of cDW. The robustness of superblack wood was superior to VACNT[12,13] and carbon aerogels superblack materials[19,20], and comparable to those of supreme black[14] and Ni-P coatings[5].

## Discussion

The natural tubular-like cellular structure of carbonized wood enables, although inefficiently, multiple internal light reflections. After modelling the optical properties of graphitic structures resembling wood, we observed that there is a limit on their blackness level that is far below that expected for superblack materials (defined as having a reflectance <0.4% in this paper). In carbonized wood, with preserved structure, the cell walls are in micrometric scale, there are several exposed fiber ends, and the wide ray cells remain intact. All of these are light-reflecting elements, thus limiting light reflectance to ca. 1–3% for carbonized wood[2,26–28], or carbonized wood modified with light-absorbing nanoparticles[29,30]. Here we fabricated superblack wood (<0.4% light reflection) by partially deconstructing these light-reflective elements and converting them into nanostructures that generate multiple internal light reflections and lower backscattering. Most of efforts on superblack materials required additional steps to create these nano-features, which is paramount to lower down reflectance. For as-grown VACNTs, multiple internal light reflections are not favored[12,13,17] unless microcavities are created between CNTs via plasma treatment[49,52]. Effective multiple internal light reflections are also essential to reach the blackness level for supreme-black[14,53,54] and nickel-phosphorus (Ni-P) superblack[55] but they usually require etching steps. Multiple internal light reflections were enabled by the nature-given tubular structure in wood, which was strengthened during the deconstruction processes that formed graded refractive index (RI) nanostructures.

Herein we used two main deconstruction strategies to lower the reflectance of carbonized wood cross-sections, i.e., (i) increased carbonization temperature, and (ii) delignification of the starting wood precursor. Starting from the carbonization temperature, we observed that increasing it from 800–1000 °C, as typically used, to over 1300 °C was sufficient to convert the microstructure of wood into a bandsaw-like microarray with graded RI and with more efficient light trapping effect. The formation of the bandsaw-like arrays occurred from the combined effect of cell deconstruction and localized lignin extrusion (Fig. 3b). The accumulation of lignin at the cell corners[40–42] created domains of high thermal recalcitrance, whereas the adjacent cells shrunk and degraded upon carbonization. Given lignin thermoplasticity[45,56], cell corners partially melted and extruded under the shrinkage stress of adjacent wood cell walls (Fig. 3b). This process enabled the creation of blacker materials (reflectance ca. 0.9%) from wood, following a route that is much simpler compared to previously reported attempts[14,15,49,57,58]. However, lowering the reflectance of carbonized wood to superblack levels required further deconstruction, via delignification, since lignin is responsible for maintaining the wood ultrastructure partially preserved. In fact, when we removed nearly all lignin in wood, a microfiber array formed from the splitting of wood cell walls during carbonization at high temperature. If not removed, residual lignin melted and penetrated through the cellulose microfibril matrix during carbonization, thus forming aggregates of microfibrils that contributed to light reflectance (Fig. 4i). A near complete delignification was required to individualize the microfibrils and to form a sparse, 100 μm tall, carbon microfiber array on the topmost (cross-section) surface of cDW (Fig. 4g, h), thus lowering light reflectance to 0.36%.

Delignified wood frozen either at −20 °C or by liquid $N_2$ before lyophilization did not show a difference on the developed superblackness (Supplementary Fig. 8 and Fig. 13). The formation of the microfiber array is specific to metal-free delignification. Carbonization (under the same conditions) of wood scaffolds delignified by $NaClO_2$ solution led to wood with a largely preserved morphology, leading to light reflectance of ~3% (Supplementary Figs. 27–29). Residual sodium species account for ca. 30 wt.% (Supplementary Fig. 28) in wood delignified by $NaClO_2$ solution, even after extended washing. Their presence as submicron particles on the surface of the wood (Supplementary Fig. 27c) increases the thermal recalcitrance of wood elements (Supplementary Fig. 27a–c) and prevents their deconstruction.

Although mild HCl acid treatment led to sodium-free wood carbon (Supplementary Fig. 27d), their light reflectance remained at ca. 3%. The results emphasized the merits of our simple metal-free delignification approach to achieve superblack properties.

Moreover, our process is not significantly affected by wood density in the range between 100 and 205 kg/m³. A higher reflectance (~0.6%) was achieved in cDW obtained from light (<90 kg/m³) and dense balsa wood (>340 kg/m³). Balsa wood led to materials with lower reflectance (~0.6%) when compared to that of pine and cedar wood-based black materials (~0.85%). Although the developed blackness (<0.85%) did not reach the level of superblack materials (defined in this paper when light reflectance <0.4%), it reached values lower than those reported for wood-based black materials (>1%)[30]. Delignification of pine and cedar wood is a challenging step given the small dimensions of their lumina, their thick cell walls, and relatively high density (>500 kg/m³). In sum, we developed sustainable superblack materials (light reflectance <0.4% and negligible light transmittance) from low-density wood. Our findings demonstrate that superblack wood absorbs up to ~99.64% of the incident light, which is the result of a low reflectivity coupled with a low transmittance. Superblack wood is fabricated from balsa wood by in-situ, metal-free delignification followed by carbonization. Wood carbonization at 1500 °C led to anatomical elements evolving into subwavelength structures that show low backscattering and enhanced light absorption. The structures formed in superblack wood led to a light reflectance value of only 0.36%, which was minimally affected by the angle of incidence/observation. The introduced superblack material outperforms the reported light-absorption performance obtained from solid wood materials and structures[2].

Superblackness expands the use of wood-based materials in light-centered technologies. Cross-sections of carbonized wood, with light absorption typically at 97%, have been used in material platforms such as solar-driven steam generation and desalination[2,26–28,30]. A material such as our superblack wood, which displays a light absorption of 99.64%, is suitable in systems such as laser measurement setups that are extremely sensitive to light reflectance. Moreover, the increased light absorption capacity of wood-based carbon should directly lead to better performance in solar-powered devices. This comes from the ability of superblack wood to harness and convert sunlight into thermal energy. For example, our superblack wood showed a stable temperature of 37.6 °C on the surface exposed to one sun (1 kW/m²) illumination. This surface temperature value is higher than those previously reported for wood-based water evaporators (30–34 °C)[26,27,30] (Supplementary Fig. 30).

Fabrication of superblack wood at large scales can be implemented using mature processing technologies associated with wood machining and carbonization. As far as material shaping, wood is conveniently transformed into designed three-dimensional (3D) macrostructures by using conventional subtractive manufacturing (e.g., carving, sculpting)[59]. Hence, the fabrication of superblack wood in complex shapes would be conveniently accomplished after the form is generated. Finally, while our method relies on the carbonization of the bulk wood (volumetric process), other methods can be tested to improve material yield and structural integrity. For instance, taking the example of flame charred surfaces[2,26–28], surface or near-surface carbonization would minimize bulk effects. Some approaches in the area point to the possibility of surface treatments to reduce the need for chemicals and solvents during processing of different wood species while generating little or no waste (Personal communication, Kenny Cheng and Prof. Philip Evans, Department of Wood Science, University of British Columbia, 2023.).

## Methods

### Materials

Balsa wood (*Ochroma pyramidale*) of 90, 105, 160, 205 and 340 mg/cm³ density were acquired from Modulor (Berlin, Germany) and used to

fabricate superblack wood. Pine wood (400 mg/cm$^3$ density) and cedar wood (560 mg/cm$^3$) density were acquired from BAUHAUS (Espoo, Finland) and Trouva (Hampshire, England). Sodium chlorite (NaClO$_2$, ≥80 wt.%), acetone (>99.5 wt.%), glacial acetic acid and hydrogen peroxide (H$_2$O$_2$) solution (35 wt.%) were supplied by Sigma-Aldrich.

## Fabrication of superblack wood

**Delignification.** Balsa wood samples were cut into 40 × 10 × 17 mm (radial × longitudinal × tangential) blocks. The samples were placed into a beaker and were separated with meshes placed along a metal-grid holder. An equal-volume mixture of hydrogen peroxide solution (35 wt.%) and glacial acetic acid was prepared[47]. The wood blocks were then infiltrated with delignification solution over-night at 25 °C under stirring. The delignification process was then carried out at 80 °C for given times. Control samples were obtained by delignification with 1 wt.% NaClO$_2$ solution at pH 4.6 (adjusted using acetic acid) at 80 °C[60]. All the delignified samples were dialyzed against de-ionized (DI) water until reaching pH 5. To retain the structural integrity, the samples were confined in a metal-grid clamp during the washing step. Wet samples were frozen at −20 °C or by liquid N$_2$ (reference sample) and freeze-dried. Stable structures were formed after lyophilization.

**Surface preparation of wood cross-section.** The cross-section surfaces of all W and DW samples were smoothed using a microtome (WSL Lab-Microtome, Switzerland). The wood samples were extracted with acetone overnight at room temperature to remove lipophilic extractives. Then the samples were washed with DI water to remove the solvent. All wood samples were frozen under −20 °C, and then the cross-section of wood was smoothed with a microtome blade until a clear and flat surface was obtained.

**Thermostabilization and carbonization.** All samples were oven-dried at 60 °C for 24 h and then subjected to thermal treatment (thermostabilization) at 250 °C for 2 h under a flowing N$_2$ atmosphere yielding oxidized samples. The heating rate was set at 1 °C/min. Carbonization was then carried out under N$_2$ flow at 800–1500 °C using 1 h holding time at a heating rate of 5 °C/min. All wood carbon was rinsed with 1 M HCl and final washing with DI water until neutral pH was reached. As a last step, all samples were dried in an oven at 105 °C.

## Characterization

**Lignin content analysis.** First, around 3 grams of wood for each delignification time were milled in a Wiley mill. Then the fraction which passed through a 30-mesh screen was collected. Then the samples were individually extracted with acetone in a Soxhlet apparatus for 6 h. Lignin content was determined in the extractives-free sample by acid hydrolysis following the NREL/TP-510-42618[61]. Briefly, each sample was hydrolyzed using sulfuric acid first at high concentration (72%) and 30 °C for 1 h, then water was added to dilute the acid to ~4% and the hydrolysis continued in an autoclave at 121 °C for 1 h. After that, the acid-insoluble lignin was obtained gravimetrically, and the acid-soluble lignin was determined in a spectrophotometer (Shimadzu UV-2550) by acquiring its absorbance at 205 nm and considering an absorptivity constant of 110 L/(g·cm). The total lignin content is the sum of the acid-soluble and -insoluble fraction. The ash content was by calcinating the oven-dried samples at 525 °C for 5 h (TAPPI 211 om-02)[62]. The lignin content was calculated on a dry basis.

**Scanning electron microscopy.** The morphology of the samples was observed by using field emission scanning electron microscopy (Zeiss Sigma VP, Germany) with an acceleration voltage of 5.0 kV. The samples were coated with a 4 nm iridium layer. All samples were first oven-dried at 90 °C for 24 h. EDS chemical semi-quantification was performed in the wood carbons obtained from both delignification methods. The EDS spectra were acquired using an Ultim® Max EDS detector from Oxford Instruments at the same acceleration voltage of 5.0 kV and a working distance of 8.5 mm.

**Raman spectroscopy.** Raman spectra were acquired in a spectrometer (Horiba LabRAM HR) equipped with a CCD camera, using an excitation laser of 633 nm. Raman was used to analyze the characteristic carbon peaks at 1350 cm$^{-1}$ and 1590 cm$^{-1}$, known as D and G bands, of carbonized wood.

**X-ray tomography.** Natural (W), delignified (DW), and carbonized delignified wood (cDW) were imaged with Xradia MicroXCT-400 device with a 1.15 μm pixel size. X-ray source voltage and power were set to 40 kV and 4 W, respectively. A total of 1861 projection images over 180° angular range (+ cone angle) were acquired with 15 s exposure time per image. The projections were reconstructed into volume images using the pi2 software (available at github.com/arttumiettinen/pi2).

**Reflectance measurement.** The spectral reflectance of the samples was measured mainly under the geometric conditions of hemispherical detection of both specular and diffuse components. A spectrophotometer (Shimadzu UV-2600, Japan) with a 60 mm-diameter integrating sphere based on barium sulfate was used for UV-Vis reflectance measurement at 8°/h geometry (8° incidence, hemispherical detection). A barium sulfate white standard was used as a reference. The integrating sphere has a 12.5 × 20 mm sample port. Detectors are placed at both the top and the bottom of the integrating sphere.

**Transmittance measurement.** A Shimadzu UV-2600 spectrophotometer with an ISR-2600 Plus integrating sphere was used for UV-Vis transmittance measurement. Samples of wood carbon with thicknesses of ~3 and 7 mm were placed in front of the beam incident port of the integrating sphere, and its transmittance was measured. The baseline signal was determined with the sample port uncovered and subtracted from the acquired sample transmittance data. The absorbance was calculated using the equation of A = 1-R-T.

**Angle varying reflectance measurement.** White light from a Halogen Light Source (Ocean Insight, HL-2000-HP-FHSA) illuminated the cross-section of carbonized wood samples in a direction with an angle θ with respect to the sample normal. The reflected light is collected at an angle θ tilted oppositely from the incident direction. Both the incident and reflected light are collected through UV-VIS collimating lens (Ocean Insight 74-UV, 200–2500 nm) connected with optical fibers (Ocean Optics, QP400-2-SR). The reflected light was guided into a spectrometer (Ocean Optics, USB2000 + UV-VIS) using an optical fiber. The reflectance spectrum was determined by dividing the observed spectrum by a diffuse reflection standard (Ocean insight WS-1, PTFE (KB)). Because the angular range of reflection is different between the sample and the reflection standard, the calculated value does not correspond to the absolute reflectance value. Thus, the spectral shape can be discussed in our results as a relative value. Reflectance spectra of various angles were measured in the detector rotation and θ−2θ scan geometry[63].

**Mechanical characterization.** Uniaxial compression was used to characterize the mechanical robustness of the wood-based superblack material. The tests were carried out parallel to the grain direction (along with tree growth direction), at a compression rate of 0.1 mm/s, using a texture analyzer (TA.XTplusC). Prior to the tests, the wood-based superblack samples were conditioned at 23 °C and 50% of relative humidity.

**Durability test**. Bare finger press test was carried out under pressure of ~7 kPa with five-times repetitions. Compressed air blow of ~50 kPa was conducted using a blow gun controlled by a pressure regulator (AR20-F02-A, SMC Pneumatics.com). Sample was tested under 10 s of air blow with five-times repetitions. Tape peeling and linear abrasion test were conducted following the durability test for superhydrophobic surface coatings[64]. Sandpaper (P2000, Mirka ECOWET) was used for linear abrasion test at a single cycle under the pressure of ~7 kPa by using ~200 g load on the area of 23 × 12 mm. Masking Tape (301 E, 3 M) was used in the tape peeling test under the pressure of ~7 kPa. The tapes were detached from the cross section of superblack wood at ~2 mm/s. The tape peeling test with conducted with five cycles.

### Application as laser beam stopper

We used an incident laser wavelength of ~515 nm with a pulse width of ~200 fs and a repetition rate of ~752 kHz, which is from an optical parametric amplifier (Light Conversion, ORPHEUS). The superblack wood sample was used as a beam stopper to block the output beam. The performance is demonstrated with a laser power ranging between ~1 and ~100 mW (measured by a power meter, S401C, Thorlabs). The transmitted power of laser through wood carbon with thicknesses of ~7 mm along the longitudinal direction is measured by the power meter behind the samples (Supplementary Fig. 24)[65]. The pulsed laser transmits through a beam splitter (50:50, BSW26R, Thorlabs, Inc.). It was noted that ~50% of incident laser transmitted through the beam splitter (50:50) while ~50% is reflected by the splitter surface. Both the beam blocks (LB1/M and LB2/M, Thorlabs, Inc.) and superblack wood were used as the laser stopper (Supplementary Fig. 24). The reflected laser (power 2 $P_1$) from the stopper surface entered the same beam splitter and ~50% of reflected laser ($P_1$) reached the power meter (PD300-3W-VI, OPHIR).

### Infrared imaging

The solar energy harvesting experiments were conducted using a multifunctional solar simulator (Abet 11002 SunLite). The illuminated light intensity was adjusted by controlling the power of the solar simulator and the distance of the optical lens. The digital multimeter (Victor VC890C+) precisely recorded the optical intensity on the wood sample. The superblack wood samples were placed in a plastic dish filled with DI water. The illuminated area on the samples was approximately 1.2 cm². Infrared (IR) images were recorded by an IR camera (PIR uc 605 infraTec) to monitor the temperature distribution of superblack wood illuminated by light at different intensities. The IR camera detected IR radiation at wavelengths from 8 to 14 μm, had a sensor resolution of 640 × 480 pixels, and was connected to a computer via Researcher software (IRBIS® 3 plus).

### Finite element simulation of optical properties

In the framework of continuum description of condensed matter, FE software COMSOL Multiphysics® 5.6.0.401 (Electromagnetic Waves, Frequency Domain in Wave Optics Module) is used for modelling the optical properties of cellular structures. In the idealized case, the cellular structures are modelled as a regular 2D array of straight rectangular (or trapezoidal) pillars with a skew bottom part (Supplementary Fig. 1). The material model used for pillars has wavelength-dependent real and imaginary parts of the refractive index (corresponding to graphite[43]). The unit cell geometry is defined by the pillar height, pillar width, distance between the pillars, and bottom slope. These geometric factors correspond to the dimension of original anatomical elements in wood carbon, e.g., fiber length, cell wall thickness, lumen width, and cell end tilt. Periodicity of the cellular structures is considered by utilizing Floquet periodic boundary conditions. Incident light (with zero incidence angle) is modelled via Port boundary condition launching a plane (electric field) wave with wavelengths in a range from 350 to 800 nm. Quadratic discretization is used for the

electric field. In total, 13 scenarios are studied. To investigate the light interactions with wood anatomical elements, 2D vertical cylinder array of straight rectangular pillars was employed in the simulation (Fig. 2e and Supplementary Fig. 2a). Light reflectance on a 2D array of straight rectangular pillars with varied fiber length (spanning 20–200 μm), cell wall thickness (spanning 0.7–6 μm), lumen width (spanning 7.5–60 μm), and cell end tilt (spanning 0–60°) are studied and shown in Supplementary Figs. 2–5. The dimensions of the anatomical elements used in the simulation were extracted from the microCT scans (Supplementary Fig. 7) and SEM images (Supplementary Figs. 15–20) of balsa wood. To examine the light interactions with formed subwavelength structures, a 2D cylinder array of trapezoidal pillars was used to replace the straight rectangular ones in the simulation (Supplementary Figs. 5–6). The dimension of the short base and the long base of the trapezoidal pillars is 0.2 μm and 1.5 μm, respectively, which is determined by the dimension of subwavelength structures in both cW (Fig. 3f) and cDW (Fig. 4h). The bandsaw-like microarrays in cW (Fig. 3d) and carbon microfiber arrays in cDW (Fig. 4g) are represented by 30 μm high (Supplementary Fig. 5a, c, e) and 100 μm high trapezoidal pillars (Supplementary Fig. 6), respectively. To create the interfiber porosity, blackbody-like cell units, in which the pillar widths are reduced to 50 nm or 5 nm, are installed adjacent to the regular cylinder array (Supplementary Fig. 6b). More details can be found in supplementary section Supplementary Figs. 1–6.

## Data availability

All data are available in the main text or the supplementary information. The raw data generated in this study have been deposited in the Figshare database at [https://doi.org/10.6084/m9.figshare.24289690].

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

## Acknowledgements

We are thankful for funding support from Commission H2020 program ERC Advanced Grant (No. 788489, BioELCell), the Canada Excellence Research Chair Program (CERC-2018-00006) and the Canada Founda-tion for Innovation (CFI Project 38623). This work was a part of the Academy of Finland's Flagship Programme under Project No. 318890 and 318891 (Competence Center for Materials Bioeconomy, FinnCERES). B.Z. is grateful for the financial support from the China Scholarship Council (Project #201702640280), the Finnish Foundation for Technol-ogy Promotion (8716), the Walter Ahlström Foundation (20230049) and the Foundation for Aalto University Science and Technology. D. Li is thanked for discussion and support with laser beam stopper measure-ments. The authors thank J. Tersteegen for assisting in the design of metal-grid clamp for wood delignification. F.X. Zou and Y.J. Dong are thanked for the support with the solar energy harvesting experiments. Vlasova M. is thanked for lignin content analysis.

## Author contributions

B.M., O.R. and A.K. idealized and supervised the project. B.Z. and B.M. designed the experiments. B.Z. conducted the experiments. X.S., Y.M. and A.K. contributed to the delignification and carbonization experiments. A.M. and T.T. performed the microCT scan and 3D recon-struction. S.K. and B.M. conducted the FE simulation. S.M. and Z.S. carried out the demonstration experiments of laser stopper. B.Z., B.M. and O.R. analyzed the data of the manuscript. B.Z. wrote the initial manuscript. All authors commented on the final manuscript.

## Competing interests
The authors declare no competing interests.
