## [Peer Review File · Nature Communications]

Reviewers' Comments:

Reviewer #1:

Remarks to the Author:

This manuscript reports on a wood-based, cost-effective & robust super black material for energy harvesting/optoelectronic technologies. The delignification of balsa wood, and following high-temperature carbonization produce 100 μm long fibrous arrays with vertically oriented cell walls where the incident light experiences multiple reflections, reducing the net reflectance to 0.36 %. The effects of carbonization temperature, wood density, and delignification process on the microscopic shape of the wood-based superblack are studied in detail, and the theoretical simulation using FEM explains the experimental data well.

Although the antireflection principle itself is simple, and there are precedents for wood black efforts, the fact that the reflectance, which in the past was $\sim 1\%$ at best, was reduced by an order of magnitude is unexpected, surprising, and commendable. The research on wood-based functional materials has been a trend in recent years, and the method used to produce wood-based superblack in this study is certainly interesting.

The manuscript is well written. The methodology is also well described with sufficient details of both manufacturing and evaluation methods, and no fatal flaws are found. Most of the claims are supported by data.

Still, I hope that the article published in Nature Communications will be on a subject that solves an important issue (or at least opens up a new avenue for it). In this research, the focus seems to be only on ease of handling and cost. A major concern is that the authors, in their superiority claim, unfairly undermine and overlook conventional superblack efforts (details to be given later). Nature Communications has a high impact, so an erroneous review of previous studies would be detrimental to a wide readership. If the goal is not only to publish this study but to pursue its practical application as a superblack material, a serious comparison should be made with the appropriate respect to the previous studies.

In short, while I appreciate the effort to develop wood super black material in this study, I cannot recommend its publication in this journal at this time without at least an improved review of previous studies. I look forward to re-reviewing the improved manuscript in light of the specific comments below as well.

Specific comments:

1) Abstract, L257-258

The authors claim that "wood density had no significant impact", but Fig. 4k and Fig. S22 show a large difference: the difference between 0.36 % and 0.64 % is only $\sim 0.3\%$, but the ratio is almost double.

2) L47, L204, L321-322, L353, L401

I can agree with the feeling that superblack should have a reflectance of $< 0.4\%$ (although the threshold setting is somewhat arbitrary), but I did not find such a definition in Ref.9. It would be more appropriate to say "In this paper, superblack is defined as having a reflectance $< 0.4\%$."

3) Introduction/Benchmarking wood-based superblack/Discussion/Supplementary discussion

The authors have relatively underestimated the previous studies by overstating the robustness, low cost, and sustainability of wood-based superblack. It needs to be rewritten. Wood-based superblack is neither the strongest material nor can be made without any treatment. Detailed comments are provided separately.

4) L54-55

The authors claim that the previous superblacks "display low cohesion and mechanically fragile", but I wonder if this is not the case for Ref.8 and Ref.9. Supreme black of Ref.8 does not withstand sandpaper but is much more durable than VANTA. The wood-based superblack in this study does not resist sandpaper too (L337-338), so if Ref.8 is considered "fragile", then the superblack of this study should also be considered fragile. A rewrite should be considered.

5) L59-60, supporting info.

As detailed in Ref. 6, the biological safety of VANTA is controversial and it may not be appropriate

to assert that it is toxic, so careful statements are necessary.

6) L63-64

The authors claim that "However, the superblackness of the latter material gradually fades under the effect of bond damage by ultraviolet (UV) radiation or atomic oxygen erosion.", but no such statement was found in Ref.8. That may be true, but if the authors wish to claim that point, the authors should show that wood-based superblack is resistant to UV and atomic oxygen in this study. If not, then deletion of this sentence would be appropriate.

Instead, I think it is fair to refer to Ref.8 in the context of:

"For a long time, nano-materials including VANTA were thought to be the key toward ultra-low reflection, but supreme black (ref.8) showed that microcavity surface textures can also be used to achieve ultra-low reflection. Microcavity textures are also present in wood-based black materials reported to date. Wood originally has a microcavity structure of cell walls. Wood-based black can be easily made by carbonization and is a sustainable material. However, it conventionally had a reflectance of more than 1%. Here we pursued a wood-based superblack material."

7) L74

The authors may have overlooked the paper by M. Zhu et al, "Tree-Inspired Design for High-Efficiency Water Extraction" (10.1002/adma.201704107), which reports 99% light absorption with wood-based black.

Interplay between light interactions and carbonized wood anatomy

8) L123-125, Fig.2d, Fig.S7

Few images show the actual cell end of the wood-based superblack; Could the authors add some SEM images or microtomographs that show the cell ends to confirm that the tilt of the cell ends is about 30 degrees?

9) L140-142

The authors claim that "The aspect ratio (H/W) of fiber length to lumen width should be > 5 to induce effective MILR and confine the incident light in the assumed cylindrical arrays (Fig. 2j, Supplementary Figs. 3i, 4 and 5f)". Where should I read that from? For example, the plot of Fig. 2j with $H/W=6.7$ shows a higher reflectance than $H/W=2.5$, and the lowest reflectance is obtained with $H/W=3.3$. A similar inversion is found in Fig. S5. This is not consistent with the authors' claim.

Fabrication of wood-based superblack material

10) Fig.2i, Fig.4c, i, k

It is not appropriate to use graphs where the vertical axis does not start from 0.

Practical implications of superblack wood.

11) L287-290

The practical implications of using only a specific beamstop product (LB2/M, Thorlabs, Inc.) as a comparison should be clarified; LB2/M seems to withstand up to 80W, but will the authors' wood-based superblack withstand this much?

12) L291

Looking at Fig.5f, the laser spot is recognizable. The statement "the laser spot on cDW nearly disappeared" is an exaggeration, is it not?

Benchmarking wood-based superblack

13) Fig.5h

Ref.5 says that Ni-P ultra-black has a reflectance of about 0.2 % (2000 ppm). Is the plot position appropriate?

14) L323

Ref.9 says that "The Nickel-Phosphorus (Ni-P) alloy is a widely accepted protective coating due to its high corrosion resistance and excellent wear resistance properties." The nickel-phosphorus (Ni-P) alloy here is NOT Ni-P superblack, but simply a protective metal coating; Ref.9 does not claim the corrosion/wear resistance of Ni-P superblack.

15) L334-337

To quantify the durability of the wood-based superblack, spectral reflectance measurements of cDW after various durability tests (compressive strength test/six-month storage/air blowing/tape peeling/bare finger touch) should be shown and compared with those before the tests.

16) L337-339

The authors state that "Sandpaper linear abrasion (P2000 under ~ 7 kPa) clearly damaged the superblackness of cDW." This is about the same level of robustness as Ref.8. What makes the distinction between "easy-to-handle" and "touch proof" in Fig.5h?

17) L339-345, and Supporting information

I think the cost calculation is unfair. The cost calculation for the wood-based superblack is based on raw material cost and utility cost, whereas the cost for other super black materials is based on the selling price of intermediate materials and/or final products and the cost of small lot outsourcing prototypes, in which case the latter is obviously more expensive.

And above all, it is overlooked that some super-black materials (black flocked fabrics) with the reflectance of 0.1-0.3 % are now available in the market.

<https://www.the-black-market.com/marketplace/mb-fabric-kiwami/>

<https://www.the-black-market.com/marketplace/ir-flock-sheet/>

The selling price of the black flocked fabrics is less than 100 euros per square meter (< 1 euro/cm²). Some black flocked fabrics are made of rayon, which can be considered a bio-derived material, and thus can be considered sustainable. The position of the wood-based superblack in this study should be properly evaluated in light of this point.

18) L409-411

The authors claim "Wood is moldable and foldable, which allows it to be further engineered into desirable shapes and geometries," but my intuition tells me that planks with the trunk section taken are more likely to split along the grain. What was the actual shear strength of the wood-based superblack?

19) Throughout Supplementary Tables

Row breaks are difficult to see.

20) Fig. S24, Tables S2-S5

The compressive strength of cDW is about 3 times different between the measured value in Fig. S24 (maximum 140 kPa) and the evaluated value in TableS5 (400 kPa). In other words, the validity of the compressive strength estimates in Tables S2-S5 seems insufficiently verified. To prevent misunderstanding, a note should be added near the compressive strength (kPa) in the tables, stating that "the compressive strength values are estimates and not actual measurements".

21) Supporting info. L482-497

It is certainly better to use fewer hazardous chemicals in production, but I think that glacial acetic acid, hydrogen peroxide, and hydrochloric acid, which are used in this study, are also hazardous. I don't think there is any need to go too far to undermine the others.

Reviewer #2:

Remarks to the Author:

The manuscript by Bin Zhao et al. describes the fabrication of wood-based superblack material through delignification and carbonization with the guidance of computational methods. This work is of high interest to fellow scholars working in multiple fields of research, including chemical engineers, material scientists, device engineers, among others. However, the following comments should be addressed before acceptance for publication.

1. For the FE simulation, what is the effect of pits in the cell wall on the light interactions?

Particularly after delignification, the pits should be opened

(<https://doi.org/10.1016/j.compositesb.2020.108296>).

2. Does the drying method of DW have an impact on the structure and properties of cDW? For example, freeze drying using liquid N₂ or supercritical CO₂ drying can even better preserve the isolated cellulose microfibrils, as compared to freezing at -20 °C followed by lyophilization.
3. Balsa wood samples of different densities were used in this study. What is the effect of wood densities on the H₂O₂/CH₃COOH delignification? Does the difference of the residual lignin content have an impact on carbonization?
4. How about using softwood for the preparation of superblack wood? Please discuss if the choice of starting wood species is important.
5. Some of the terms in the manuscript should be better defined for broader readership. For example, line 76, "micro-channeled architecture" is not a good term for wood. Line 77-78, better to define the size of "tubular morphologies" and "fibrillar arrays". Line 83-84, line 208, line 230, "cellulose fibrils", "fibril assembly", "wood fibrils", "fibril bundle", these terms should be better defined. Line 87, "nano-microfiber arrays", what is nano-microfiber?
6. Fig. 3c, better use tangential, radial, longitudinal coordinate system instead of x, y, z. Some photographs of samples are missing scale bar or sample geometry in the figure legend.
7. As indicated by the authors, superblack materials is important for energy harvesting and optoelectronic technologies. Will the superblack wood fabricated in this work improve the performance of solar evaporation/steam generation reported in references 2, 17, 18?

Reviewer #3:

Remarks to the Author:

The authors use computational methods to design superblack wood, an effort to develop antireflective materials that are more mechanically robust than other alternatives. This is a very nice paper that will make a wonderful contribution to the literature, and I am excited to follow the further development of super black wood for diverse applications. I was particularly happy to see the authors test the robustness of superblack wood under several test regimes. My only broad comment is that the authors do not engage with the substantial literature on biological super blacks, ranging from bird of paradise feathers to peacock spiders and deep sea fish and beetles. It may be worth adding a few sentences to compare their results to other biological materials that have been previously described. I leave the choice of whether or not to do so up to the authors. Thank you for the opportunity to read this interesting paper!

Minor comments:

Introduction: acronyms make it harder for readers to understand work. I suggest you remove all acronyms that are not in extremely common usage (e.g., DNA is easy to understand; MILR is not easy to understand despite the fact that I research them.) Replace all MILR, CNT, and OBG

57-69: Here, you do not reference the substantial existing ways that nature has developed super black materials—e.g., in peacock spiders, bird feathers, and deep sea fish. It may be worth referencing some of this work given that the focus of your paper is mechanical toughness (something these natural superblacks achieve).

91: In Figure 1, add scale bars. Excellent figure!

115-127: very nice results

140-143: the ratio of fiber length to width discussion herein, which is fascinating, reminds me of the strange feather adaptations in birds of paradise which have an extremely deep cavity with quite narrow walls (in this case, composed by long curving feather barbules). Similarly, the bandsaw-like tips in Figure 3 call to mind the sub-wavelength prongs, or barbule extensions, on bird of paradise feathers

184-190: do the bandsaw arrays act like a graded refractive index?

277: Here, you discuss oblique light which is very important— for some of the super black animals

described, they also reflect far more light at oblique angles due to Fresnel surface reflection (just like VANTA black).

Figure 5 is an excellent experimental test of robustness and angle-independence.

408-409: not true if you include naturally occurring organisms to be biobased materials

RESPONSE TO REVIEWERS' COMMENTS

We acknowledge the opinion and suggestions provided by the expert referees of our manuscript (NCOMMS-23-35727), titled “*Wood-based Superblack*”. All the issues raised are addressed in the revised version of the manuscript and the Supporting Information document. The itemized list below incorporates all the comments and our point-by-point reply. In the revised manuscript, the changes are highlighted with given font effects and color associated to each of the reviewers (see MARKED version of the manuscript). We use **blue**, **green** and **red** to highlight the changes made according to the comments provided by reviewer **#1**, **#2** and **#3**, respectively. We appreciate very much the suggestions and insightful feedback provided by the expert reviewers. They have helped to enhance the quality, breadth, and impact of our contribution.

REVIEWER #1:

General Comment. This manuscript reports on a wood-based, cost-effective & robust super black material for energy harvesting/optoelectronic technologies. The delignification of balsa wood and following high-temperature carbonization produce 100 μm long fibrous arrays with vertically oriented cell walls where the incident light experiences multiple reflections, reducing the net reflectance to 0.36 %. The effects of carbonization temperature, wood density, and delignification process on the microscopic shape of the wood-based superblack are studied in detail, and the theoretical simulation using FEM explains the experimental data well. Although the antireflection principle itself is simple, and there are precedents for wood black efforts, the fact that the reflectance, which in the past was ~ 1 % at best, was reduced by an order of magnitude is unexpected, surprising, and commendable. The research on wood-based functional materials has been a trend in recent years, and the method used to produce wood-based superblack in this study is certainly interesting. The manuscript is well written. The methodology is also well described with sufficient details of both manufacturing and evaluation methods, and no fatal flaws are found. Most of the claims are supported by data. Still, I hope that the article published in Nature Communications will be on a subject that solves an important issue (or at least opens up a new avenue for it). In this research, the focus seems to be only on ease of handling and cost. A major concern is that the authors, in their superiority claim, unfairly undermine and overlook conventional superblack efforts (details to be given later). Nature Communications has a high impact, so an erroneous review of previous studies would be detrimental to a wide readership. If the goal is not only to publish this study but to pursue its practical application as a superblack material, a serious comparison should be made with the appropriate respect to the previous studies. In short, while I appreciate the effort to develop wood super black material in this study, I cannot recommend its publication in this journal at this time without at least an improved review of previous studies. I look forward to re-reviewing the improved manuscript in light of the specific comments below as well.

General answer. *We would like to thank the reviewer for recognizing the quality of our work and its relevance. Our effort is centered on developing new subwavelength carbon structures directly from wood, which are highly efficient in absorbing light in the visible range. We demonstrate the performance of the wood-based black material using a laser measurement setup. The developed material adds to (does not compete with) current efforts in the area, for example microcavity supreme black and carbon nanotube arrays. In the revised version of the manuscript, we incorporate the results of a thorough review of previous studies and those reported during the time the elapsed since the submission of our manuscript. To our knowledge, our measured light reflectivity remains as the lowest reported so far for wood-based materials but we fully recognize*

the past and ongoing research efforts to convert wood into black materials. In an attempt to fairly relate our results with the ever evolving scientific findings, we retract any claim on performance superiority or ranking. All the specific comments have been addressed as shown below. Please see the MARKED version of the revised manuscript and the Supplementary Information document.

Comment 1. Abstract, L257-258. The authors claim that "wood density had no significant impact", but Fig. 4k and Fig. S22 show a large difference: the difference between 0.36 % and 0.64 % is only ~0.3 %, but the ratio is almost double.

→ **Authors 1.** *Thanks for the constructive comment. A minor role can be assigned to the density (in the range of our study, between 90 and 340 kg/m³) in relation to the structural development. However, the dimensions of the wood-derived black materials depend on the density, affecting the reflectivity. The effect of wood density on the blackness is now more clearly discussed in the revised manuscript.*

Comment 2. L47, L204, L321-322, L353, L401. I can agree with the feeling that superblack should have a reflectance of <0.4 % (although the threshold setting is somewhat arbitrary), but I did not find such a definition in Ref.9. It would be more appropriate to say "In this paper, superblack is defined as having a reflectance <0.4 %."

→ **Authors 2.** *We fully agree that the differentiation between black and superblack is arbitrary. The revised manuscript refers to the definition of superblack as having a total reflectance <0.4 % and added a clarifying note to the text.*

Comment 3. Introduction/Benchmarking wood-based superblack/Discussion/Supplementary discussion. The authors have relatively underestimated the previous studies by overstating the robustness, low cost, and sustainability of wood-based superblack. It needs to be rewritten. Wood-based superblack is neither the strongest material nor can be made without any treatment. Detailed comments are provided separately.

→ **Authors 3.** *We appreciate the opportunity to qualify our materials more properly. We have reworded the introduction and discussion according to the comment provided by the reviewer. We also remove the discussion, mostly qualitative and relative in its nature, associated with cost, hazard and sustainability impacts. Finally, we provide a more comprehensive assessment of the mechanical integrity of our material compared to those reported in the literature.*

Comment 4. L54-55 The authors claim that the previous superblacks "display low cohesion and mechanically fragile", but I wonder if this is not the case for Ref.8 and Ref.9. Supreme black of Ref.8 does not withstand sandpaper but is much more durable than VANTA. The wood-based superblack in this study does not resist sandpaper too (L337-338), so if Ref.8 is considered "fragile", then the superblack of this study should also be considered fragile. A rewrite should be considered.

→ **Authors 4.** *We agree with the reviewer regarding our overlook and unsubstantiated comparison of superblack materials. In the revised version of the manuscript, we avoid such comparison and rather focus on our discoveries on the development of a superblack material from wood. We highlight the uniqueness of our developed materials, which is framed as complementary to VANTA and microcavity materials.*

Comment 5. L59-60, supporting info. As detailed in Ref. 6, the biological safety of VANTA is controversial and it may not be appropriate to assert that it is toxic, so careful statements are necessary.

→ **Authors 5.** *Thanks for the suggestion. Although it is known that CNTs are toxic towards some tissues and organisms (10.1021/ar300028m), we concur that CNT toxicity when assembled in structures such as VANTA has not been investigated in detail. Therefore, we refrain from stating any possible biological and toxicological effects associated with VANTA.*

Comment 6. L63-64: The authors claim that "However, the superblackness of the latter material gradually fades under the effect of bond damage by ultraviolet (UV) radiation or atomic oxygen erosion.", but no such statement was found in Ref.8. That may be true, but if the authors wish to claim that point, the authors should show that wood-based superblack is resistant to UV and atomic oxygen in this study. If not, then deletion of this sentence would be appropriate. Instead, I think it is fair to refer to Ref.8 in the context of: "For a long time, nano-materials including VANTA were thought to be the key toward ultra-low reflection, but supreme black (ref.8) showed that microcavity surface textures can also be used to achieve ultra-low reflection. Microcavity textures are also present in wood-based black materials reported to date. Wood originally has a microcavity structure of cell walls. Wood-based black can be easily made by carbonization and is a sustainable material. However, it conventionally had a reflectance of more than 1%. Here we pursued a wood-based superblack material."

→ **Authors 6.** *Thanks for the fair criticism and detailed justification. We agree to remove the sentence related to the damage by ultraviolet (UV) radiation or atomic oxygen erosion for Supreme black. Moreover, we agree with the assessment and suggestion provided by the reviewer and have added text to better introduce each major effort related to superblack materials. The latter include Ni-P, VANTA, and microcavity supreme black. The revised text refers to results reflected in each of the referenced contributions. We removed non-substantiated claims of degradation even if it was possible that polymers such as those used in supreme black were subjected to such condition under UV light.*

Comment 7. L74. The authors may have overlooked the paper by M. Zhu et al, "Tree-Inspired Design for High-Efficiency Water Extraction" (10.1002/adma.201704107), which reports 99% light absorption with wood-based black.

→ **Authors 7.** *Thanks for the reference. The above publication by M. Zhu et al. has been added to the manuscript along with other ones.*

Comment 8. Interplay between light interactions and carbonized wood anatomy L123-125, Fig.2d, Fig.S7 Few images show the actual cell end of the wood-based superblack; Could the authors add some SEM images or microtomographs that show the cell ends to confirm that the tilt of the cell ends is about 30 degrees?

→ **Authors 8.** *Thanks for the suggestion. We wish to clarify that the FE simulation showed that an increased tilt of the cell ends provided a greater path-length for light propagation and, hence, enhanced light absorption, for instance compared to that of lumina with blunt ends. The simulation helped us to evaluate how "imperfections" of wood affect light absorption, contrarily to "empty tubes". With FE we showed that light reflectance drops as the cell tilt angle is increased (up to 30°). It is known that the angle of cell end tilt for balsa wood vary widely, from 10 to 80°. Hence, the inclusion of micrographs showing a given tilt of the cell ends can be misinterpreted. We clarify*

that we used a tilt of 30 ° only as a reference for investigating other parameters; the implications of the tilt of the cell ends are discussed more thoroughly in the revised manuscript.

Comment 9. L140-142. The authors claim that "The aspect ratio (H/W) of fiber length to lumen width should be > 5 to induce effective MILR and confine the incident light in the assumed cylindrical arrays (Fig. 2j, Supplementary Figs. 3i, 4 and 5f)". Where should I read that from? For example, the plot of Fig. 2j with H/W=6.7 shows a higher reflectance than H/W=2.5, and the lowest reflectance is obtained with H/W=3.3. A similar inversion is found in Fig. S5. This is not consistent with the authors' claim.

→ **Authors 9.** *We agree with the assessment provided by the reviewer. To avoid any ambiguity and not to mislead the reader, we modified the graph to display the fiber length instead of the aspect ratio. The fiber length should be $> 50 \mu\text{m}$ to induce effective multiple internal light reflections and to confine the incident light in the assumed cylindrical arrays.*

Comment 10. Fabrication of wood-based superblack material. Fig.2i, Fig.4c, i, k It is not appropriate to use graphs where the vertical axis does not start from 0.

→ **Authors 10.** *The vertical axes in Figures 2i, Fig.4c,i,k have been updated and now start from 0.*

Comment 11. L287-290 Practical implications of superblack wood. The practical implications of using only a specific beamstop product (LB2/M, Thorlabs, Inc.) as a comparison should be clarified; LB2/M seems to withstand up to 80W, but will the authors' wood-based superblack withstand this much?

→ **Authors 11.** *Thanks for the constructive comment. We tested another beam stopper product (LB1/M, Thorlabs, Inc.) and the results are now added (**Supplementary Table 1**). In brief, under strong laser illumination (100 mW) only ~ 30 ppm of scattered reflection was observed with our superblack wood. This scattered reflection is lower than that for laser beam block LB1/M, and 3 times lower than laser beam block LB2/M. Although LB2/M can withstand up to 80W, we compared our superblack wood with commercial beam stops using a typical laser illumination power range, 1 to 100 mW. We have added discussions in the main text and Supplementary information about the subject.*

Comment 12. L291: Looking at Fig.5f, the laser spot is recognizable. The statement "the laser spot on cDW nearly disappeared" is an exaggeration, is it not?

→ **Authors 12.** *We agree. The above claim has been deleted from the manuscript.*

Comment 13. Benchmarking wood-based superblack. Fig.5h Ref.5 says that Ni–P ultra-black has a reflectance of about 0.2 % (2000 ppm). Is the plot position appropriate?

→ **Authors 13.** *Thanks for the constructive comment. Following the previous comments, we have decided to remove the plot that compares superblack materials. We, still keep the discussions on mechanical integrity but following with substantial modifications accounting for all major classes of superblack (see "practical implications of superblack wood").*

Comment 14. L323. Ref.9 says that "The Nickel–Phosphorus (Ni–P) alloy is a widely accepted protective coating due to its high corrosion resistance and excellent wear resistance properties."

The nickel–phosphorus (Ni–P) alloy here is NOT Ni–P superblack, but simply a protective metal coating; Ref.9 does not claim the corrosion/wear resistance of Ni–P superblack.

→ **Authors 14.** *Thanks for the constructive comment. See Authors 13.*

Comment 15. L334-337. To quantify the durability of the wood-based superblack, spectral reflectance measurements of cDW after various durability tests (compressive strength test/six-month storage/air blowing/tape peeling/bare finger touch) should be shown and compared with those before the tests.

→ **Authors 15.** *The reflectance spectra of superblack wood before the durability tests and after six-month storage, air blowing, tape peeling and bare finger touch test are now added to the revised Supplementary Information document, Figure 25. Mild water washing (no stirring) and oven drying at 105 °C were used as finishing step during the fabrication routine of superblack wood. The compressive strength and sandpaper linear abrasion test compromised the blackness of superblack wood. These aspects are now included in the discussion of the revised manuscript.*

Comment 16. L337-339. The authors state that "Sandpaper linear abrasion (P2000 under ~7 kPa) clearly damaged the superblackness of cDW." This is about the same level of robustness as Ref.8. What makes the distinction between "easy-to-handle" and "touch proof" in Fig.5h?

→ **Authors 16.** *See Authors 13 and 15. The superblack and microcavity materials appear to display approximately the same level of mechanical integrity compared to our system. Accordingly, the discussion has been revised.*

Comment 17. L339-345, and Supporting information. I think the cost calculation is unfair. The cost calculation for the wood-based superblack is based on raw material cost and utility cost, whereas the cost for other super black materials is based on the selling price of intermediate materials and/or final products and the cost of small lot outsourcing prototypes, in which case the latter is obviously more expensive. And above all, it is overlooked that some super-black materials (black flocked fabrics) with the reflectance of 0.1-0.3 % are now available in the market.

<https://www.the-black-market.com/marketplace/mb-fabric-kiwami/>

<https://www.the-black-market.com/marketplace/ir-flock-sheet/>

The selling price of the black flocked fabrics is less than 100 euros per square meter (<1 euro/cm²). Some black flocked fabrics are made of rayon, which can be considered a bio-derived material, and thus can be considered sustainable. The position of the wood-based superblack in this study should be properly evaluated in light of this point.

→ **Authors 17.** *We believe that this manuscript brings several unique aspects related to materials development. However, we acknowledge that our initial attempt to assess the cost was qualitative and not well supported. On that note, all the quantitative data related to cost calculation have been removed to avoid unfair comparisons based on the cost of raw materials and market/product prices. In the revised version of the manuscript, we now add the superblack flocked fabrics available in the market, a true research-to-business success story that deserves attention.*

Comment 18. L409-411

The authors claim "Wood is moldable and foldable, which allows it to be further engineered into desirable shapes and geometries," but my intuition tells me that planks with the trunk section taken are more likely to split along the grain. What was the actual shear strength of the wood-based superblack?

→ **Authors 18.** *The reviewer raises a relevant question. We reformulated our sentence for precision. “Wood can be sculpted into three-dimensional (3D) shapes using conventional subtractive manufacturing (e.g., carving, sculpting), which can be applied as means to develop superblack wood with complex shapes.”*

Comment 19. Throughout Supplementary Tables Row breaks are difficult to see.

→ **Authors 19.** *Row breaks are added to all Supplementary Tables.*

Comment 20. Fig. S24, Tables S2-S5

The compressive strength of cDW is about 3 times different between the measured value in Fig. S24 (maximum 140 kPa) and the evaluated value in Table S5 (400 kPa). In other words, the validity of the compressive strength estimates in Tables S2-S5 seems insufficiently verified. To prevent misunderstanding, a note should be added near the compressive strength (kPa) in the tables, stating that "the compressive strength values are estimates and not actual measurements".

→ **Authors 20.** *We removed the discussion about estimated compressive strength values in Tables S2-S5, which were qualitative. For carbonized wood (cD) and carbonized delignified wood (cDW), the compressive strength values reflect the measured data displayed in Fig. S25.*

Comment 21. Supporting info. L482-497 It is certainly better to use fewer hazardous chemicals in production, but I think that glacial acetic acid, hydrogen peroxide, and hydrochloric acid, which are used in this study, are also hazardous. I don't think there is any need to go too far to undermine the others.

→ **Authors 21.** *We removed discussion about safety, toxicity and cost estimation, as reflected in the comments above. To highlight the innovation of superblack wood, the low hazard impact associated with superblack wood production is briefly mentioned.*

REVIEWER #2

General Comment. The manuscript by Bin Zhao et al. describes the fabrication of wood-based superblack material through delignification and carbonization with the guidance of computational methods. This work is of high interest to fellow scholars working in multiple fields of research, including chemical engineers, material scientists, device engineers, among others. However, the following comments should be addressed before acceptance for publication.

General Answer. *We appreciate the encouraging comments provided by the reviewer and opinion about innovativeness and quality. Please see the MARKED version of the revised manuscript and the Supplementary Information document.*

Comment 1. For the FE simulation, what is the effect of pits in the cell wall on the light interactions? Particularly after delignification, the pits should be opened (<https://doi.org/10.1016/j.compositesb.2020.108296>).

→ **Authors 1.** *This is an interesting question. The revised version of the manuscript contains a discussion about the possible effect of pits on the light absorption capacity of superblack wood.*

Comment 2. Does the drying method of DW have an impact on the structure and properties of cDW? For example, freeze drying using liquid N₂ or supercritical CO₂ drying can even better

preserve the isolated cellulose microfibrils, as compared to freezing at -20 °C followed by lyophilization.

→ **Authors 2.** *Delignified wood frozen at -20 °C and at lower temperature (liquid N₂) maintained the cellular structure with loosened cellulosic fibers after lyophilization. Although delignified wood frozen by liquid N₂ better preserved the isolated cellulose microfibrils, it did not show a difference on the generated microfiber arrays and the developed superblackness. During the carbonization, shrinkage stress split cell walls into smaller entities (the carbon microfiber arrays in this work). Two factors matter the most: (1) the carbonization temperature (1500 °C) leading to the splitting of cell walls and the (2) complete removal of lignin loosening the cellulose microfibrils in the cell wall. The revised manuscript highlight related aspects.*

Comment 3. Balsa wood samples of different densities were used in this study. What is the effect of wood densities on the H₂O₂/CH₃COOH delignification? Does the difference of the residual lignin content have an impact on carbonization?

→ **Authors 3.** *From our experience, it is more challenging to remove lignin from denser wood. The delignification step takes longer time for dense balsa due to their smaller cell lumina, thicker cell walls and greater lignin content. We have added discussion in the revised manuscript covering such subjects. In short, balsa of medium density (100-250 kg/m³) took 4 h for full delignification while it took 6 h for denser balsa (> 340 kg/m³) and 2 h for light balsa (<90 kg/m³). Considering the high carbon yield of lignin (45 wt.%) compared to that of cellulose (20 wt.%), it is expected that residual lignin affects the carbon yield of delignified wood. More importantly, residual lignin limits the cell wall disintegration during carbonization due to their higher adhesion and thermal recalcitrance. The revised manuscript incorporates this discussion in detail.*

Comment 4. How about using softwood for the preparation of superblack wood? Please discuss if the choice of starting wood species is important.

→ **Authors 4.** *The main requirement to achieve superblack wood is a starting material with a low density or low filling factor. This justified our choice of low-density balsa, which is much lighter than most softwoods and other hardwoods. Nevertheless, we confirm that similar carbon microfiber arrays was formed with other wood species, provided that they have similar density or cell dimensions. We experimentally demonstrate this using delignified pine and cedar. The carbon microfiber arrays were formed in pine but not entirely in cedar, which led to light reflectance of 0.77% and 0.85 %, respectively. We have included discussions about this in the revised manuscript.*

Comment 5. Some of the terms in the manuscript should be better defined for broader readership. For example, line 76, “micro-channeled architecture” is not a good term for wood. Line 77-78, better to define the size of “tubular morphologies” and “fibrillar arrays”. Line 83-84, line 208, line 230, “cellulose fibrils”, “fibril assembly”, “wood fibrils”, “fibril bundle”, these terms should be better defined. Line 87, “nano-microfiber arrays”, what is nano-microfiber?

→ **Authors 5.** *We agree with the suggestions provided by the reviewer and to refer to wood anatomical elements using generic terms for the benefit of the broad readership. We have made several modifications along the manuscript aiming at clarifying some few terms, and to keep a consistent nomenclature.*

Comment 6. Fig. 3c, better use tangential, radial, longitudinal coordinate system instead of x, y, z. Some photographs of samples are missing scale bar or sample geometry in the figure legend.

→ **Authors 6.** *We now make reference to the tangential, radial, and longitudinal coordinate system for the wood structure (Fig. 3c-3g). Scale bars are added to all photographs of samples.*

Comment 7. As indicated by the authors, superblack materials is important for energy harvesting and optoelectronic technologies. Will the superblack wood fabricated in this work improve the performance of solar evaporation/steam generation reported in references 2, 17, 18?

→ **Authors 7.** *The efficiency of solar evaporators is correlated with the light absorption capacity of the material, which in turn tracks with equilibrium temperature of the materials when exposed to sunlight. We have added a set of experiments to exemplify the possible gains of using superblack materials in a solar evaporator system.*

REVIEWER #3

General Comment. The authors use computational methods to design superblack wood, an effort to develop antireflective materials that are more mechanically robust than other alternatives. This is a very nice paper that will make a wonderful contribution to the literature, and I am excited to follow the further development of super black wood for diverse applications. I was particularly happy to see the authors test the robustness of superblack wood under several test regimes. My only broad comment is that the authors do not engage with the substantial literature on biological super blacks, ranging from bird of paradise feathers to peacock spiders and deep-sea fish and beetles. It may be worth adding a few sentences to compare their results to other biological materials that have been previously described. I leave the choice of whether or not to do so up to the authors. Thank you for the opportunity to read this interesting paper!

General Answer. *We appreciate the kind opinion expressed by the reviewer. The mechanical robustness of superblack materials is particularly important for practical applications. Biological super blacks definitely provide inspiration to the fabrication approaches and we now make mention to such possibility along with revisions according to the comments below. **Please see the MARKED version of the revised manuscript and the Supplementary Information document.***

Comment 1. Introduction: acronyms make it harder for readers to understand work. I suggest you remove all acronyms that are not in extremely common usage (e.g., DNA is easy to understand; MILR is not easy to understand despite the fact that I research them.) Replace all MILR, CNT, and OBG.

→ **Authors 1.** *We have replaced all MILR, OBG and CNT by their full names. The latter is used repeatedly and we make more frequent use of the abbreviation. In all cases we refer to the full name of the abbreviation when first mentioned.*

Comment 2. 57-69: Here, you do not reference the substantial existing ways that nature has developed super black material-e.g., in peacock spiders, bird feathers, and deep-sea fish. It may be worth referencing some of this work given that the focus of your paper is mechanical toughness (something these natural superblacks achieve).

→ **Authors 2.** *We have included several biological super black materials found in nature (see the revised discussion and we modified the text). For instance, superblack scales in deep-sea fish are developed by using melanosomes with optimized size and aspect ratio, which may inspire the design of robust synthetic superblack materials.*

Comment 3. 91: In Figure 1, add scale bars. Excellent figure! 115-127: very nice results
→ *Authors 3.* We thank the reviewer for the comment. Scale bars are added to all photographs.

Comment 4. 140-143: the ratio of fiber length to width discussion herein, which is fascinating, reminds me of the strange feather adaptations in birds of paradise which have an extremely deep cavity with quite narrow walls (In this case, composed by long curving feather barbules). Similarly, the bandsaw-like tips in Figure 3 call to mind the sub-wavelength prongs, or barbule extensions, on bird of paradise feathers. 184-190: do the bandsaw arrays act like a graded refractive index?
→ *Authors 4.* Compared to flat cell wall in wood, the generated bandsaw arrays showed graded dimensionality, from bandsaw tips toward the bottom. The bandsaw arrays act like a graded refractive index, where the refractive index jumps are reduced so as the light scattering. This leads to a much better explanation for reflectance drop in wood carbon with the bandsaw arrays. We have added new discussions in the text to further emphasize such aspects.

Comment 5. 277: Here, you discuss oblique light which is very important- for some of the super black animals described, they also reflect far more light at oblique angles due to Fresnel surface reflection (just like VANTA black).
→ *Authors 5.* The reviewer raises an interesting point. The blackness of our superblack wood does not change much when increasing the observation angles up to 55 ° from the sample normal. Fresnel surface reflection in VANTA black is reported but it is much lower in our superblack wood.

Comment 6. Figure 5 is an excellent experimental test of robustness and angle-independence.
→ *Authors 6.* Thanks for the comment. Superblack materials are usually synthesized by materials at low-filling fraction, which typically raise the issue of mechanical integrity. Therefore, we feel that it is necessary to evaluate the blackness relative to the robustness of the material, in the same figure. For completeness, we now add a figure displaying the light reflectance of superblack wood following wear testing to the supplementary information, which in fact shows a remarkable integrity of our superblack wood compared to other carbon-based superblack materials.

Comment 7. 408-409: not true if you include naturally occurring organisms to be biobased materials.
→ *Authors 7.* We have included discussion about several biological super black systems. Superblack wood emerges as a man-made material based on a biological structure. The respective sentence has been revised in the main manuscript.

Reviewers' Comments:

Reviewer #1:

Remarks to the Author:

I appreciate that the authors have carefully and thoroughly revised the manuscript. Now I believe the reviewers' comments have been adequately addressed. I am pleased to recommend this work for publication in Nature Communications as it is.

Reviewer #2:

Remarks to the Author:

I would like to thank the authors for addressing my comments.

Reviewer #3:

Remarks to the Author:

The revised manuscript is interesting and well-edited. I have no further comments! Congratulations to the authors on some fine work.

RESPONSE TO REVIEWERS' COMMENTS

We acknowledge the opinion and suggestions provided by the expert referees of our manuscript (NCOMMS-23-35727A), titled “*Wood-based Superblack*”. All the issues raised are addressed in the revised version of the manuscript and the Supporting Information document. The itemized list below incorporates all the comments and our point-by-point reply. There were no required changes in the main manuscript and supplementary information.

REVIEWER #1:

General Comment. I appreciate that the authors have carefully and thoroughly revised the manuscript. Now I believe the reviewers' comments have been adequately addressed. I am pleased to recommend this work for publication in Nature Communications as it is.

General answer. *We would like to thank the reviewer for recognizing the quality of our work and its relevance to the readership of Nature Communications.*

REVIEWER #2

General Comment. I would like to thank the authors for addressing my comments.

General Answer. *We appreciate the encouraging comments provided by the reviewer and opinion about innovativeness and quality. We thank again for the opportunity that was given to us to further discuss additional aspects of our contribution.*

REVIEWER #3

General Comment. The revised manuscript is interesting and well-edited. I have no further comments! Congratulations to the authors on some fine work.

General Answer. *We appreciate the kind opinion expressed by the reviewer.*